# OpenReview forum: "MAGIC: A Co-Evolving Attacker–Defender Adversarial Game for Robust LLM Safety"
_ICML.cc/2026/Conference — ICML 2026 regular_

### Official Review · Reviewer_uJcj · 2026-03-06

**Soundness:** 2
**Presentation:** 3
**Significance:** 3
**Originality:** 3
**Overall Recommendation:** 4
**Confidence:** 2

**Summary:**

This paper proposes MAGIC, a multi-round, multi-agent RL framework that models the secure alignment of LLM as an asymmetric sequential game. The framework decouples the attacker and defender to mitigate optimization conflicts. It first initializes the attacker's reasoning with SFT, and then conducts GRPO. The experimental results demonstrate improvements across multiple benchmarks.

**Compliance With Llm Reviewing Policy:**

Affirmed.

**Key Questions For Authors:**

Beside the weaknesses mentioned above, a few more questions:

1.In Table6, why is the KL coefficient is set to 0?

2.In Table2, please explain the performance drop in AlpacaEval 2 and MMLU for Qwen2.5-14B.

3.Given the reliance on alternating GRPO, what evidence can be provided to support that the proposed method approach the SPNE described in Theorem 3.2?

**Limitations:**

yes

**Strengths And Weaknesses:**

Strengths:

1.The idea of constructing safe alignment as an asymmetric order game is novel.

2.The paper evaluates the method on various security benchmarks, including single-round and multi-round jailbreak settings.

Weaknesses:

1.The authors claim guaranteed pointwise safety based on SPNE. However, the method relies on alternating GRPO updates. In high-dimensional, non-convex RL, alternating optimization does not guarantee convergence to SPNE.

2.The rewards are obtained by a fixed third-party model (Qwen3Guard), which can easily cause the defender to overfit to the judge's blind spots rather than learning intrinsic safety.

3.The ablation study shows that the SFT initialization has strong influence, which contradicts the claim that the framework autonomously discovers vulnerabilities through RL.

4. Although the authors mention other recent multi-agent safety research (e.g. AdvEvo-MARL), no comparisons are provided in the experimental section.

---

> ### Author Rebuttal · Authors · 2026-03-30
>
> We thank the reviewer for the constructive comments and hope the following clarifications address the concerns. If there are any remaining questions, we would be happy to provide more details.
> ### **W1 & Q3: About SPNE convergence**
> Due to character limits, we provide more analysis and empirical evidence of convergence in responses to Reviewer HH2Z's W1 and TNUX's Q2.
>
> ### **W2: MAGIC framework is not tied to particular reward model**
> We provide detailed experimental results in the responses to Reviewer g3tx's W1.
>
> ### **W3: Concern that the performance gain may come from SFT initialization**
> The main performance gains come from the co-evolutionary RL training rather than the specific distillation source. Please refer to the responses to Reviewer g3tx's W2.
>
> ### **W4: Comparison with multi-agent safety research (e.g. AdvEvo-MARL)**
> We thank the reviewer for pointing out AdvEvo-MARL as a relevant recent work. While both AdvEvo-MARL and our method involve adversarial co-evolution, they target fundamentally different problem settings and address different research questions. AdvEvo-MARL studies **safety in multi-agent systems (MAS)**, where multiple agents interact with each other and safety failures can arise from agent manipulation, corrupted inter-agent communication, or hijacked user instructions. Its goal is to internalize safety into task agents so that the **entire multi-agent system** remains robust during collaborative interaction.
>
> By contrast, our work focuses on a **single defender model** facing adversarially rewritten prompts from an attacker. More specifically, the two works differ along three key dimensions. First, the **system setting** is different: AdvEvo-MARL considers a partially observable Markov game with multiple attackers and defenders interacting over a full MAS trajectory, whereas our work studies robustness of one defender model against evolving prompt attacks.  Second, the **threat model** is different: AdvEvo-MARL evaluates threats such as agent manipulation, message corruption, and user instruction hijacking in chain, tree, and fully connected topologies, while our setting centers on adversarial prompt rewriting and jailbreak robustness for a single model.  Third, the **training objective and evaluation target** are different: AdvEvo-MARL jointly optimizes defenders for both task performance and system-level safety in MAS, whereas our goal is to improve the defender’s refusal robustness while preserving benign compliance and general instruction-following ability.
>
> For these reasons, we believe a direct empirical comparison would be difficult to interpret fairly, since the two methods are designed for different environments, attack surfaces, and evaluation goals. We will revise the paper to make this distinction more explicit and cite AdvEvo-MARL in the discussion of related co-evolutionary safety training.
>
> ### **Q1: why is the KL coefficient is set to 0?**
> We intentionally set the KL coefficient to 0 to avoid keeping the attacker overly close to the initial policy or constraining it to the original 20 mutation-based rewriting patterns Instead, our goal is to encourage broader exploration so that the attacker can discover more diverse attack strategies during co-evolution.
>
> In addition, this choice is also broadly consistent with recent follow-up work on GRPO-style / reasoning-oriented RL, where the KL term is often removed in practice. For example, DAPO[1] explicitly excludes the KL term for long-CoT RL, arguing that this restriction is unnecessary in that setting; [2] reports strong results without any KL regularization; and [3] also removes the KL term and sets $\beta = 0$. We will include these details in the revised paper for transparency.
>
> ### **Q2: Concern about the performance drop of Qwen2.5-14B in Tab. 2**
> We thank the reviewer for pointing this out, and we also noticed this performance drop. As discussed in Reviewer HH2Z's W3, we believe this reflects the inherent safety–utility trade-off, rather than a contradiction in our method. The purpose of our study is not to show that the full MAGIC setup achieves the best result on every individual metric, but to illustrate how each component affects the overall balance between safety and utility.
>
> In fact, a similar phenomenon was also observed in the baseline Self-RedTeam. To address utility degradation, Self-RedTeam mixes in SFT updates on a self-distilled dataset $D_{\mathrm{SFT}}$ concurrently with $L_\text{RL}$. By contrast, our current MAGIC results are obtained with RL training alone, without such additional SFT mixing. We will clarify this point in the revision.
>
> **Reference**
>
> [1] Yu et al. "Dapo: An open-source llm reinforcement learning system at scale." arXiv:2503.14476.
>
> [2] Hu et al. "Open-reasoner-zero: An open source approach to scaling up reinforcement learning on the base model." arXiv:2503.24290.
>
> [3] Liu et al. "Understanding r1-zero-like training: A critical perspective." arXiv:2503.20783.

---

> > ### Author Rebuttal · Reviewer_uJcj · 2026-04-04
> >
> > The rebuttal from the authors have resolved my concerns (though some questions may need further justifications). I tend to keep my rating.

---

> > > ### Author Response · Authors · 2026-04-04
> > >
> > > We sincerely appreciate your careful re-evaluation and are glad that our rebuttal has addressed your main concerns.
> > > We are truly grateful for your positive feedback and recognition of our work.
> > > If you have any remaining points that require further clarification or more detailed justifications, we would be more than happy to provide additional explanations in the revision.

---

### Official Review · Reviewer_g3tx · 2026-03-07

**Soundness:** 3
**Presentation:** 2
**Significance:** 3
**Originality:** 3
**Overall Recommendation:** 4
**Confidence:** 3

**Summary:**

This paper proposes MAGIC, a multi-turn multi-agent reinforcement learning framework for LLM safety alignment. The framework models safety as an asymmetric adversarial sequential game between an attacker and a defender. The attacker is first warm-started via SFT on CoT-enriched data, and then both agents are iteratively co-evolved using GRPO. This paper then launches experiments spanning multiple model families (Qwen2.5, Llama3.1) across safety, benign compliance, and general capability benchmarks.

**Compliance With Llm Reviewing Policy:**

Affirmed.

**Final Justification:**

The independent judge evaluation (W1), distillation source ablation (W2), and re-run of Self-RedTeam (W3) collectively address my main concerns. I raise my score accordingly. The decision is weak accept.

**Key Questions For Authors:**

1. Can you report the main results (Table 1) using an independent judge model that was not used during training? This would show whether the safety gains hold beyond Qwen3Guard-specific overfitting.
2. What happens if the CoT completions in Phase 1 are generated by an open-source model instead of Gemini-2.5-Pro? An ablation on the distillation source would help isolate MAGIC's own contribution.
3. Can you re-run Self-RedTeam under the same experimental settings as MAGIC (same judge model, decoding parameters, and benchmark splits) for a fair comparison on the Llama3.1 family?
4. Table 5 and Equation 6 overflow the margins. Please reorganize the layout to comply with ICML formatting requirements.

**Limitations:**

yes

**Strengths And Weaknesses:**

## Strengths

1. **Good problem motivation and formulation.** The asymmetric sequential game formulation is a meaningful improvement over symmetric self-play approaches (e.g., Self-RedTeam).
2. **Comprehensive experimental coverage.** The evaluation is extensive, spanning 9 safety benchmarks, and 5 general capability benchmarks. And the paper provides comprehensive ablation studies.
3. **Reproducibility effort.** The paper provides detailed hyperparameters, prompt templates, and classification standards.


## Weaknesses

1. **Same model (Qwen3Guard) for both training and evaluation.** Qwen3Guard is used as the reward model during GRPO training (Section 4.2) and as the safety judge during evaluation. This means the defender is explicitly optimized to satisfy the same classifier that later evaluates it.
2. Phase 1 uses Gemini-2.5-Pro to generate CoT completions for all SorryBench attack strategies. The quality of the attacker's initial reasoning, is therefore largely inherited from a single model. This raises a concern: the contribution of the CoT-enriched Attack Pool Benchmark is hard to disentangle from the capability of the distillation source.
3. **The comparison with Self-RedTeam is not fair.** For Llama3.1, the authors "directly report the results from the original paper without re-running the experiments" , meaning Self-RedTeam and MAGIC do not share the same experimental settings.

---

> ### Author Rebuttal · Authors · 2026-03-30
>
> We thank the reviewer for the constructive comments and hope the following clarifications address the concerns. If there are any remaining questions, we would be happy to provide more details.
> ### **W1 & Q1: The model remains effective under other safety reward models.**
> At present, to the best of our knowledge, only WildGuard and Qwen3Guard provide explicit refusal-related metrics for adversarial benign prompts, which makes them the most suitable choices for evaluating over-refusal in this setting. To further address the reviewer’s concern about reward-model-specific optimization, we additionally report evaluation results using WildGuard and GPT-4o as alternative safety judges. **The experimental results indicate that MAGIC framework is not tied to any particular reward model**. Due to the character limit, we provide additional WildGuard evaluation results at: https://anonymous.4open.science/r/Anonymous-icml2026-D737/eval_wildguard.png.
>
> **GPT-4o as judge model**
> | Method | WG:Test AH↓| WG:Test AH↓ | WJB AH↓ | DAN ↓ | HB AH↓ | HB VH↓ | OR-Bench VH↑ | XSTest VH↑ | StrongREJECT VH↑ | WJB AB↑ | XSTest VB↑ |
> |---|---:|---:|---:|---:|---:|---:|---:|---:|---:|---:|---:|
> | *Qwen2.5-7B-Instruct* |0.537 |0.049 |0.656 |0.357 |0.253|0.405 |0.902 |0.91 |0.964|0.996|0.904|
> | + Self-RedTeam |0.386  |0.017 | 0.401|0.283 |0.283| 0.156 |0.979 |0.965 |0.988 |0.996 |0.860 |
> | + **MAGIC(ours)** |0.214 |0.019 |0.262 |0.083 |0.171| 0.122 |0.971 | 0.970|0.988 |0.988 |0.876 |
> | *Qwen2.5-14B-Instruct* |0.398 |0.032 |0.484 |0.227 |0.217|0.097  |0.922 |0.940 | 0.978|1.000 |0.904 |
> | + Self-RedTeam |0.220 |0.012 |0.292|0.143 |0.114 |0.047 |0.980 |0.965 | 0.990|0.984 |0.900 |
> | + **MAGIC(ours)** |0.101 |0.007 | 0.084|0.023 |0.061 |0.028 |0.995 |0.985 | 0.994|0.968 |0.868 |
> | *Llama3.1-8B-Instruct* |0.466 |0.090 |0.529 |0.493 |0.402|0.288  |0.874 |0.950 |0.983 |0.988 |0.900 |
> | + **MAGIC(ours)** |0.089 |0.022 |0.054 |0.060 |0.128 |0.192 |0.956 |0.975 |0.989 |0.964 |0.904 |
> ### **W2 & Q2: Concern about performance gain is from CoT completions**
> In fact, the model trained with SFT only does not exhibit strong attack performance. The SFT stage mainly provides the attacker with an initial reasoning process, rather than endowing it with strong attack capability. This can also be observed from the ASR gap between **attack-SFT** and **no-revision** in Fig.3 (**21.25 vs 20.00**), which suggests that SFT mainly serves as an initialization step rather than the main source of the final performance gain.
>
> To address the reviewer’s concern, **we further replace Gemini-2.5-Pro with Qwen2.5-72B-Instruct as the distillation source in Phase 1 and observe the same overall conclusion**: MAGIC consistently outperforms both the base model and Self-RedTeam across multiple safety benchmarks, while largely preserving benign helpfulness and instruction-following performance. We acknowledge that a stronger distillation source can provide a better initialization and thus improve performance to some extent. This is also consistent with our analysis in the paper that **the main performance gains come from the subsequent co-evolutionary RL training rather than the specific distillation source itself**.
>
> | Method | WG:Test AH↓| WG:Test AH↓| WJB AH↓| DAN ↓| HB AH↓| HB VH↓| OR-Bench VH↑| XSTest VH↑| StrongREJECT VH↑| WJB AB↑| XSTest VB↑|
> |---|---:|---:|---:|---:|---:|---:|---:|---:|---:|---:|---:|
> | *Qwen2.5-7B-Instruct* |0.365| 0.038| 0.701| 0.327| 0.363| 0.250| 0.892 |0.800| 0.964| 0.992| 0.940|
> | + Self-RedTeam |0.255| 0.017 |0.442| 0.323 |0.237| 0.047| 0.973| 0.825| 0.988| 0.980| 0.904|
> | + w/ Qwen2.5-72B-Instruct | 0.068 |0.000 |0.267 |0.097 |0.124 |0.078  |0.988| 0.865|0.997 | 0.980| 0.900|
> | + w/ Gemini-2.5-pro |0.023 |0.002| 0.198| 0.043| 0.055| 0.019| 0.977| 0.860| 0.988| 0.968| 0.945 |
> ### **W3 & Q3: Re-run Self-Redteam for Llama 3.1**
> For a fair comparison, we re-ran Self-RedTeam with the authors’ open-source code under the same settings as MAGIC. On Llama3.1, the two methods achieve similarly strong safety, though Self-RedTeam is slightly better on some harmful-refusal metrics. However, it causes a much larger drop in benign compliance, especially on WJB adv benign (**0.512 vs. 0.960**) and AlpacaEval 2 (**27.8% vs 33.2%**), while MAGIC preserves benign behavior much better.
> | Method | WG:Test AH↓| WG:Test AH↓ | DAN ↓ | HB AH↓ | HB VH↓ | OR-Bench VH↑ | XSTest VH↑ | StrongREJECT VH↑ | WJB AB↑ | XSTest VB↑ | AlpacaEval2 ↑|
> |---|---:|---:|---:|---:|---:|---:|---:|---:|---:|---:|---:|
> | *Llama3.1-8B-Instruct* |0.279 |0.075  |0.466 |0.324 |0.266 |0.890 |0.930 |0.983 |0.988 |0.912 |33.7% |
> | + Self-RedTeam |0.006 |0.007 |0.008 |0.057 |0.031|0.966 | 0.945 |0.982 |0.512 |0.888 |27.8% |
> | + **MAGIC(ours)** |0.017 |0.017 |0.007 |0.143 |0.072 |0.928 |0.930 |0.986 |0.960 |0.924 | 33.2%|
> ### **Q4: Formatting requirements**
> We reorganize the layout of Tab. 5 and Eq. 6 in the revised version to ensure full compliance with the ICML formatting guidelines.

---

> > ### Author Rebuttal · Reviewer_g3tx · 2026-04-02
> >
> > I thank the authors for the detailed rebuttal and additional experiments. My concerns are well addressed.
> >
> > Therefore, I raise my score accordingly from 3 to 4.

---

> > > ### Author Response · Authors · 2026-04-03
> > >
> > > We sincerely appreciate your encouraging follow-up and are very glad to hear that our rebuttal helped clarify the paper. We are also truly grateful for your positive assessment and for recognizing the value of our clarifications.

---

### Official Review · Reviewer_TNUX · 2026-03-10

**Soundness:** 3
**Presentation:** 3
**Significance:** 3
**Originality:** 2
**Overall Recommendation:** 4
**Confidence:** 2

**Summary:**

This paper proposes MAGIC, a framework that trains an attacker and a defender in a co-evolving process to improve LLM safety. The attacker rewrites harmful queries into more subtle jailbreak prompts, while the defender learns to detect and refuse them without harming normal responses. Through alternating reinforcement learning, both sides gradually improve. Experiments show that this approach significantly reduces attack success rates while largely preserving the model’s general capabilities.

**Compliance With Llm Reviewing Policy:**

Affirmed.

**Final Justification:**

weak accepted

**Key Questions For Authors:**

1. Is there evidence of reward hacking in the proposed training process? The reward model employed in this work is Qwen3Guard. Based on my prior experimental experience, this model is not particularly strong and can be relatively easy to identify as operating under a safety-oriented reward pattern. Consequently, it raises the question of whether policies optimized against this reward model would generalize to other reward models. In particular, would the learned behavior remain effective when evaluated with alternative safety models such as LlamaGuard, or with larger and more capable reward models?
2. The paper formulates the problem as a sequential game and discusses concepts such as SPNE. However, the actual training procedure is implemented as an alternating RL–based approximate optimization, which does not guarantee convergence to any game-theoretic equilibrium. As a result, the theoretical analysis appears to function more as an interpretive framework rather than providing a rigorous guarantee of the algorithm’s properties. It would therefore be helpful if the authors could provide theoretical convergence analysis or empirical evidence supporting the stability or convergence behavior of the proposed training process.
3. The authors state "the attacker’s ever-changing strategies continuously uncover long-tail vulnerabilities" and further analyze it in Q5. However, the attacker initialization relies on a constructed Attack Pool based on template mutation, which effectively constrains the attack exploration space from the outset. If the initial attack distribution lacks sufficient diversity, the co-evolution process may simply recombine existing patterns rather than discovering genuinely new attack paradigms. Consequently, drawing such a conclusion from a limited number of cases does not appear to provide sufficiently solid evidence.
4. How does the training computational cost of this method compare with that of existing approaches?
5. I did not clearly understand what the core difference is between MAGIC and prior work [1][2] that jointly trains attackers and defenders through co-evolutionary exploration. If this distinction cannot be clearly articulated, the novelty of MAGIC may appear limited.

[1] Toward Optimal LLM Alignments Using Two-Player Games. EMNLP2025 Findings. \
[2] Adversarial Attack-Defense Co-Evolution for LLM Safety Alignment via Tree-Group Dual-Aware Search and Optimization

The points above summarize my main concerns and questions. If any of them stem from a misunderstanding of the paper, I would appreciate the authors’ clarification. Should these issues be adequately addressed, I would be willing to reconsider and adjust my evaluation of this work.

**Limitations:**

yes

**Strengths And Weaknesses:**

### Strengths
The paper frames jailbreak defense as a dynamic adversarial game rather than a static red-teaming problem. Modeling the interaction between attacker and defender as a sequential game is intuitive and aligns well with how real-world attacks evolve. MAGIC introduces a clean co-evolution training pipeline where an attacker generates increasingly sophisticated jailbreak prompts while the defender learns to reject them. Decoupling the attacker and defender models avoids optimization conflicts that may arise in self-red-teaming approaches.
### Weaknesses
See the key questions

---

> ### Author Rebuttal · Authors · 2026-03-30
>
> We thank the reviewer for the constructive comments and hope the following clarifications address the concerns.
> ### **Q1: MAGIC framework is not tied to particular reward model**
> Due to character limits, we provide detailed experimental results in the responses to Reviewer g3tx's W1.
>
> ### **Q2: Theoretical convergence analysis or empirical evidence of convergence**
> Our analysis is explanatory and does not claim convergence of Algorithm 1. Theorem 3.2 only shows that any SPNE solution satisfies the safety constraints. Existing convergence results for alternating RL-based optimization, e.g., [1,2], apply only to simultaneous-move games. And they rely on strong assumptions such as strong convex-concavity, positive lower bounds on policy probabilities, and finite action spaces, which do not hold in our LLM-based sequential game. To analyze convergence, we also explored the two-timescale approach in [3], but they also require technical assumptions that are difficult to justify here. We therefore do not claim a strict guarantee, and instead use [3] only as motivation for adjusting $T_D$ and $T_A$. Please see our response to Reviewer HH2Z's W1&Q1, where we explain in detail the motivation behind our training method. We also provide the experimental results for training curves which shows stable optimization without collapse ( https://anonymous.4open.science/r/Anonymous-icml2026-D737/ablation.png).
>
> ### **Q3: Evidence for uncovering long-tail vulnerabilities beyond the initial pool**
> We agree that the exploration boundary of any automated red-teaming framework is shaped by initialization, and MAGIC is no exception. Our claim is therefore not that MAGIC removes initialization dependence, but that it changes how exploration proceeds after initialization. Unlike prior automated red-teaming methods that mainly rely on heuristic mutation or iterative rewriting, MAGIC enables the attacker to learn through interaction with the defender how to combine strategies effectively. This makes exploration more adaptive and less reliant on manually designed expansion rules.
>
> The representative case in Q5, together with Tab. 16, clearly demonstrates the emergence of previously unseen combinatorial attack patterns discovered during co-evolution, highlighting our method’s ability to uncover long-tail vulnerabilities beyond the initial pool. **This interpretation is further supported by the bottom panel of Fig. 4**, which shows that strategies that are rare or underexplored in the SFT data become more represented after RL training, including multi-condition stacking and complex logic nesting.
>
> ### **Q4: Training computational cost**
> Our training pipeline consists of two stages: SFT and RL. In our current setup, when using Qwen2.5-7B-Instruct as the base model, the SFT stage takes approximately 3 hours on 4 H200 GPUs, while the RL stage takes approximately 13 hours on 4 H200 GPUs.
>
> ### **Q5: Comparsion with prior work**
> Our novelty lies not in co-evolution alone, but in three specific design choices: the asymmetric sequential game formulation, the co-evolutionary RL procedure, and the attacker initialization. We will cite these work in the revision and provide a more detailed discussion with MAGIC.
>
> Compared with [2], MAGIC differs in three main ways: it formulates jailbreak-defense as an **asymmetric sequential game** targeting **SPNE** rather than a zero-sum game under **Nash equilibrium**, making the defender a pointwise best response to each attack; it is naturally compatible with **multi-turn jailbreaks** through sequential attacker-defender interaction; and it addresses attacker cold start with a **CoT-enriched Attack Pool**, enabling policy-level strategy composition through RL rather than relying mainly on diversity rewards or explicit search over predefined templates.
>
> Compared with [4], which is a concurrent work, the difference is mainly methodological. ACE-Safety is fundamentally tree-search-centric: it relies on GS-MCTS over a predefined structured strategy space, together with AC-TGPO for training. In contrast, MAGIC does not rely on explicit tree search. Instead, it learns a reasoning-capable attacker policy through CoT initialization + RL co-evolution under an asymmetric sequential game formulation. As a result, ACE-Safety mainly composes attacks through search over strategy templates, while MAGIC focuses on policy-level compositional exploration through iterative learning.
>
> **Reference**
>
> [1] Cen et al. "Fast policy extragradient methods for competitive games with entropy regularization." NeurIPS 2021.
>
> [2] Zheng et al. "Toward optimal llm alignments using two-player games." EMNLP2025 Findings.
>
> [3] Hong et al. "A two-timescale stochastic algorithm framework for bilevel optimization: Complexity analysis and application to actor-critic." SIAM Journal on Optimization, 2023.
>
> [4] Li et al. "Adversarial Attack-Defense Co-Evolution for LLM Safety Alignment via Tree-Group Dual-Aware Search and Optimization." arXiv.

---

> > ### Author Rebuttal · Reviewer_TNUX · 2026-04-03
> >
> > Thank you for your response. While your rebuttal has addressed some of my concerns, the main issues I raised—particularly Q1 and Q3—remain insufficiently resolved, as the core aspects of these questions were not fully addressed. Nevertheless, I appreciate your efforts in clarifying several points, and I am willing to increase my score from 3 to 4.

---

> > > ### Author Response · Authors · 2026-04-03
> > >
> > > We sincerely appreciate your encouraging follow-up and are very glad to hear that our rebuttal helped clarify the paper. We are also truly grateful for your positive assessment and for recognizing the value of our clarifications.
> > > In addition, it is also possible that our understanding of the core aspects of Q1 and Q3 does not fully align with your concerns; we would be happy to provide a more detailed discussion in our revision if you be willing to share further guidence.

---

### Official Review · Reviewer_HH2Z · 2026-03-13

**Soundness:** 3
**Presentation:** 4
**Significance:** 4
**Originality:** 4
**Overall Recommendation:** 5
**Confidence:** 3

**Summary:**

This paper presents MAGIC, an iterative co-evolution strategy that models adversarial prompt injection as a turn-wise game between attacker and defender. SPNE is used as the theoretical basis to model the game, which helps provide safety guarantees. Results show that co-evolution outpaces self-red-teaming in generating more robust defenders.

**Compliance With Llm Reviewing Policy:**

Affirmed.

**Final Justification:**

The main concern about the disconnect between the SPNE theoretical basis and practical implementation in the paper was adequately addressed in the rebuttal, albeit with authors promising to revise the final paper. In accordance, I have changed my score from Weak Accept to Accept.

**Key Questions For Authors:**

How do you justify the mismatch as explained in Weakness 1? Without a nested loop optimization, how do you ensure that the co-evolution will converge towards the SPNE equilibrium $\pi^*_D$?

**Limitations:**

yes

**Strengths And Weaknesses:**

## Strenghts
- The problem formulation is novel and the incorporation of SPNE is innovative
- The co-evolution framework is effective not only for safety training but could be extended to other adversarial training regimes to develop novel behaviors
- The evaluation is elaborate and support the paper's claims and the research questions in section 5.3 are meaningful

## Weaknesses
- There seems to be a mismatch in Eq.1 and Eq.2 and the iterative optimization described in Section 4, particularly Algorithm 1. The double expectation structure of Eq.2 would imply that the optimization would involve a double nested loop, however that is not the case.
- While the observation of compositional attack strategies is interesting as an emergence of this co-evolution, stating it as a contribution is not adequately justified as a pointed prompt to an LLM to combine different adversarial techniques might more reliably produce the same results as exemplified in Page 8 Line 397
- The ablation studies do not consistently demonstrate strong results for the full MAGIC setup, weakening the justification for several design choices

---

> ### Author Rebuttal · Authors · 2026-03-30
>
> We thank the reviewer for the constructive comments and hope the following clarifications address the concerns. If there are any remaining questions, we would be happy to provide more details.
> ### **W1 & Q1: Mismatch between the double expectation in Eq. 2 and the lack of nested loops in Algorithm 1.**
>
> The double expectation structure of Eq.2 implies that finding the SPNE is a bilevel optimization problem. Solving this bilevel optimization problem is difficult, especially in the context of language models. However, there are some techniques from classic theoretical work that can be drawn upon, such as the two-timescale algorithm used in [1]. Specifically, the learning rate $\beta_k$ used for the inner optimization and the learning rate $\alpha_k$ used for the outer optimization satisfy $\lim_{k \to \infty}\frac{\alpha_k}{\beta_k}=0$. With this special choice of learning rates, together with a series of assumptions required for convergence, it can be proven that the alternating iterations converge to the optimal solution of the bilevel optimization problem.
>
> Returning to our LLM attack-defense setting, it is first worth clarifying that the assumptions required for the proof in [1] do not hold in our case. For example, it assumes that the objective function is twice differentiable, the gradient is Lipschitz continuous, and that the objective function is strongly convex, all of which are difficult to satisfy in the context of LLMs. As a result, we did not claim  in the paper that this algorithm can theoretically converge to an SPNE, we  encourage convergence in practice by introducing two different update scales. In Algorithm 1, this can be implemented by adjusting $T_D$ and $T_A$ in Lines 7 and 15, respectively. The roles of $T_D$ and $T_A$ are analogous to those of learning rates. Empirically, we found that maintaining them at the same scale leads to better convergence behavior.
>
> ### **W2: Combining different adversarial techniques might more reliably produce compositional attack strategies.**
>
> We agree that a carefully designed prompt could construct a specific compositional attack instance. However, we believe this is different from the core contribution of our work. Our goal is not to demonstrate the mere existence of a particular compositional attack, but to propose a co-evolutionary RL framework in which the attacker **can autonomously learn which strategies to combine, how to combine them, and when to combine them** through interaction with the defender. It highlights the broader potential of using LLMs directly as attackers for red teaming.
>
> Additionally, when the initial attack space becomes large, manual composition can hardly cover **semantically meaningful** higher-order combinations. Their scalability is inherently limited, since the combinatorial space grows rapidly and human-designed rules or search methods become increasingly inefficient in exploring diverse yet meaningful compositions. In contrast, MAGIC delegates this exploration process to the attacker policy itself and continuously reinforces effective patterns through feedback. Therefore, the example on Page 8 should not be viewed as an isolated claim, but rather as a qualitative illustration of the exploratory capability enabled by our framework. It suggests that, under co-evolution, the attacker is not simply repeating initial templates, but is gradually developing more effective attack patterns through feedback.
>
> ### **W3: Concern about the consistency of the ablation results.**
>
> We agree that the ablation results do not show consistent dominance across all metrics, which we attribute to **the inherent trade-off between safety and utility** [2]: improving one aspect often comes at the expense of another. Therefore, the purpose of the ablation study is not to show that the full MAGIC setup is best on every individual metric, but to illustrate how each component affects the overall safety–utility trade-off.
>
> The results in Tab.5 also show a consistent pattern. For instance, No-Game performs well on XSTest and AlpacaEval, but its robustness on adversarial benign inputs drops substantially (**0.496 vs. 0.968** for full MAGIC). Defender-only and MAGIC-base also achieve strong refusal rates on harmful prompts, but they noticeably hurt utility and instruction-following performance (e.g., AlpacaEval **23.491 / 28.184 vs. 33.224** for full MAGIC). In contrast, full MAGIC significantly improves robustness against harmful prompts over the base model while largely preserving benign helpfulness and instruction-following performance.
>
> **Reference**
>
> [1] Hong et al. "A two-timescale stochastic algorithm framework for bilevel optimization: Complexity analysis and application to actor-critic." SIAM Journal on Optimization, 2023.
>
> [2] Anthropic. "Training a helpful and harmless assistant with reinforcement learning from human feedback." arXiv:2204.05862.

---

> > ### Author Rebuttal · Reviewer_HH2Z · 2026-04-05
> >
> > Thank you for your rebuttal. My concerns in W2 and W3 have been answered.
> >
> > Regarding W1, the details provided in the rebuttal clarify the paper's method and must be included in future revisions of the paper.
> >
> > The clarification unfortunately brings forward concerns that there is a disconnect between the SPNE theoretical basis and the paper's implementation. I acknowledge that performing a bilevel optimization with LLMs is intractable, thus the approximation of alternative optimization is necessary. However, the paper lacks mention of this factor entirely, and does not reference Algorithm 1, $T_D$, or $T_A$ in the text. The rebuttal mentions "...maintaining them at the same scale leads to better convergence behavior", which goes against the explanation that the two-timescale approach will approximate a bilevel optimization. Additionally, learning rates would be applicable at the GRPO updates (i.e., lines 12 and 20 of Algorithm 1), so how exactly do $T_D$ and $T_A$ play the role of learning rates?

---

> > > ### Author Response · Authors · 2026-04-07
> > >
> > > We thank the reviewer for the helpful follow-up and for recognizing our clarifications regarding W2 and W3. To address the remaining concerns in W1, we will make more explicit in the revision how Algorithm 1 serves as a practical approximation to the SPNE formulation. Specifically, Eqs. (1)–(2) define the target equilibrium concept, while Algorithm 1 is a practical alternating approximation motivated by the same bilevel/Stackelberg structure, rather than an exact nested-loop solver. We will also explain how this approximation is implemented through $T_D$, $T_A$, and the alternating update schedule.
> > >
> > > - Why we use $T_D$ and $T_A$ to control the dynamics, rather than GRPO learning rates.
> > >
> > > We use $T_D$ and $T_A$ because they control the relative adaptation speed of the two players at the algorithmic level, playing a role analogous to learning rates in two-timescale optimization. Specifically, the GRPO learning rate controls the magnitude of each individual parameter update, whereas $T_D$ and $T_A$ control how many consecutive updates are allocated to the defender and attacker before switching. Therefore, although $T_D$ and $T_A$ are not optimizer step sizes, they determine how quickly one player adapts relative to the other during alternating optimization, i.e., how closely one player can track the other. This design is also consistent with recent work such as [1] and [2], where two-timescale behavior is realized through asymmetric inner and outer update counts. For example, in [1] the ratio of inner to outer iterations is $K_t$, and in [2] the ratio is $T-1$.
> > >
> > > Another practical reason is that, in full-parameter fine-tuning of LLMs, the training process is highly sensitive to the learning rate: a relatively large learning rate may cause forgetting and poorer generalization, whereas a small one may slow convergence and weaken the model’s ability to learn new tasks. In practice, the learning rate for RL fine-tuning of 7B models is typically in the range of $5\times10^{-7}$ to $3\times10^{-6}$ [3–4]. However, the result in [5] theoretically requires the learning rate to satisfy $\alpha_k \ll \beta_k$, which is difficult to realize within such a narrow and already conservative range. For these reasons, we choose to control the two-timescale dynamics through the numbers of inner and outer update steps, rather than through markedly different GRPO learning rates.
> > >
> > > - Why our main experiments keep $T_D$ and $T_A$ on the same scale.
> > >
> > > The key reason lies in practical considerations: in our setting, the attacker is harder to train well than the defender. The main challenge on the attacker side is not only optimization, but also exploration: the attacker must discover and maintain a diverse set of effective and long-tail attack strategies. By contrast, once informative attacks are available, the defender’s alignment objective is relatively easier, since it mainly needs to adapt to the current attack distribution. As a result, making the defender much faster than the attacker can be counterproductive: the defender may overfit to the current attack distribution before the attacker has sufficiently expanded it, thereby weakening the intended co-evolution. For this reason, our main setting keeps $T_D$ and $T_A$ at comparable scales as a stability–exploration trade-off: the defender still adapts to the attacker, while the attacker retains enough update budget to continue exploring diverse strategies.
> > >
> > >
> > > To further support this point empirically, we will add reward-curve comparisons under different update ratios and GRPO learning-rate settings in the revision. This ablation is intended to show exactly the trade-off discussed above: increasing defender-side update asymmetry does not improve co-evolution stability in our setting, because the bottleneck lies more in maintaining sufficient attacker exploration than in making the defender track faster.
> > >
> > > **Reference**
> > >
> > > [1] Online bilevel optimization: Regret analysis of online alternating gradient methods, AISTATS 2024
> > >
> > > [2] Tighter analysis of alternating stochastic gradient method for stochastic nested problems. NeurIPS 2021.
> > >
> > > [3] Deepseekmath: Pushing the limits of mathematical reasoning in open language models. arXiv:2402.03300.
> > >
> > > [4] Open-reasoner-zero: An open source approach to scaling up reinforcement learning on the base model. arXiv:2503.24290.
> > >
> > > [5] A two-timescale stochastic algorithm framework for bilevel optimization: Complexity analysis and application to actor-critic. SIAM Journal on Optimization, 2023.

---

### Decision · Program_Chairs · 2026-04-30

**Decision:**

Accept (regular)

**Comment:**

The recommendation is based on the reviewers' comments, the area chair's evaluation, and the author-reviewer discussion. This paper studies LLM safety by formulating safety alignment learning as a co-evolving attacker–defender adversarial game. All reviewers find the studied setting novel and the results provide new insights. The authors’ rebuttal has successfully addressed the major concerns of reviewers. In the post-rebuttal phase, all reviewers were satisfied with the authors’ responses and agreed on the decision of acceptance.
Overall, I recommend acceptance of this submission. I also expect the authors to include the new results and suggested changes during the rebuttal phase in the final version.

Additional note of automatically flagged references -- Reference: Lab, S. A. Openrt: An open-source red teaming framework for multimodal llms. arXiv preprint arXiv:2601.01592, 2026.

Issue: authors mismatch with arXiv

AC: Please correct the authors names to match the arxiv version, instead of using the Lab as authors